# Biomechanical Analysis of Patient-Specific Temporomandibular Joint Implant and Comparison with Natural Intact Jaw Bone Using Finite Element Method

Anubhav Tiwari [1,*], Vijay Kumar Gupta [1], Rakesh Kumar Haldkar [2] and Ivan A. Parinov [2]

1   Discipline of Mechanical Engineering, Machine Dynamics and Vibrations Laboratory, PDPM Indian Institute of Information Technology Design & Manufacturing Jabalpur Dumna, Airport Road, Jabalpur 482005, India; vkgupta@iiitdmj.ac.in

2   I. I. Vorovich Mathematics, Mechanics and Computer Sciences Institute, Southern Federal University, 344090 Rostov-on-Don, Russia; rakeshhaldkar@gmail.com (R.K.H.); parinov_ia@mail.ru (I.A.P.)

*   Correspondence: 1713605@iiitdmj.ac.in

**Abstract:** The purpose of this study is to design a patient-specific TMJ implant and study its behaviour under different loading conditions compared with natural intact TMJ. There are several diseases, which affect the proper growth and function of TMJ, and in some cases, TMJ injury results from accidents. To repair the TMJ, temporomandibular joint replacement or TJR surgery is performed. In this work, CT-scan data of the skull and mandible region with broken condylar head were used to study the biomechanical behaviour of the intact mandible and customized TMJ prostheses in order to design a patient-specific total TMJ implant. The customized TMJ implant was virtually studied under simulated loading conditions using finite element method (FEM) in ANSYS Workbench and then compared to the intact jaw-mandible for the combinations of two different biocompatible material models. It is observed that the natural TMJ has a higher deformation value as compared to the patient-specific TMJ implant due to the lower mechanical strength of bone relative to the Ti-6Al-4V and Co-Cr alloy. Hence, we can conclude that the designed custom TMJ implant is safe for the patient from the point of design perspective.

**Keywords:** patient-specific TMJ implant; TMD; FEM; stress shielding

## 1. Introduction

The joint that connects the temporal bone (skull) with the mandible (jaw) is commonly known as the temporomandibular joint (TMJ). There are two TMJs, one on each side, where they provide motion to the jaw. The muscles attached near the TMJ allow the motion of opening and closing of the mouth. This joint is very intricate and could be categorized as a kind of synovial joint that acts in a spherical joint manner making it a complicated structure [1]. Complexity in TMJ is due to the fact that it consists of a sliding hinge that allows the jaw to move up and down and also enables side to side, and back to front motion. This joint consists not only of bone and muscle but also a small piece of cartilage that acts as a dampener and protects the brain from direct shock and the bones from wear and tear in the place of motion.

However, in some cases, the joints may move out of the regular path and not move in the way it is supposed to, giving rise to a problem. This problem or dysfunction is commonly known as Temporomandibular disorder (TMD). In most cases, in adults from the age of 20 to 40 years, up to 15% have TMD with more in females compared to males [2]. TMD is a disease that could be diagnosed by plentiful indications such as aching in the joint region and mandible, limited opening of the oral cavity, pain in case of opening of the jaw, changes in biting force and pain with chewing food or malfunction of the mandible. TMD hence causes weakening of the joint in many activities such as chewing, speaking, and

yawning and may lead to terrible failure of TMJ [3]. Most TMD consists of osteoarthritis, TMJ disk degeneration, condylar fracture (irreversible), and resorption of the condylar head due to uncertain reasons.

To reinstate the normal functioning of the TMJ and to re-establish the normal structure and form, a custom-made total joint implant of TMJ is required [4]. This custom-made TMJ implant has many more advantages when compared to endogenous joint replacement. The better functionality of the TMJ implant is in its enhanced stability, which gives a more normal structure and form, provides a good aesthetic appearance, and is comparatively more suitable, however, it has a few drawbacks such as decrease of intraoperative time and needs for biocompatibility with human tissue/body [5]. To understand these complex problems, we cannot just rely on one surgical/medical discipline, but also require the knowledge of multiple areas such as mechanical and biologic material properties, and design, so as to increase the effectiveness of the implant design. Moreover, with help of the mechanical domain, we can account for the data related to the bone and the stress acting on it. From material properties, we can understand that the implant material such as screws or plates or bars are usually 10 to 20 times more rigid than bone. Few researchers have worked on the examination of mandibular biomechanics. The loading conditions after the implant placement should be studied so as to avoid fractures in the bone or having the stress shielding effect.

Currently, the leading manufacturers of the TMJ implant are TMJ Concept (they manufacture the patient-specific implant) and Biomet (they manufacture the ready-to-use implant or stock implant) [6]. As per one of the surveys, it was found that every year there are as many as 1000 cases on average of total alloplastic TMJ reconstruction surgery that has been performed in the USA alone which is of a population of 328 million people [6]. We can concur that it makes this the proper time for India to take it up along with the oral and maxillofacial surgeons to develop an implant that suits the Indian form and structure along with biomedical scientists and bio-engineers. We need to concentrate on TMJ disorders as they affect nearly 5% of the population and require proper diagnosis and help. The implant is mostly designed considering the geometry and anatomy of the Americans or European peoples and as we know they are different considering the Indian anatomy and form making it more suitable for the Indian patient.

From the above scenario, we can understand that the current circumstances necessitate an immediate proposal and design, development of an Indian patient-specific implant and ready to use implant (stock implant) for people suffering from TMD and then generate the data for future use. The design of the TMJ implant accompanies many problems such as load distribution and transfer within the bodies i.e., between fossa and condyle, and in the case of an implant with screws too. We need to cut down or diminish the stress shielding process as it leads to a reduction in bone density and to achieve this, we need to transfer the load more towards the condyle region from the lateral side of the bone (at the fixing screw region). Another problem is the variation in kinematics during changing the normal rotation of these joints or restricting lateral movements. As a researcher in this article, we discuss a patient-specific TMJ implant and propose the design and development of the same. The developed implant has two major components i.e., the fossa part and ramus part. In this study, two material models were used separately. In one material model, Ultra-High Molecular Weight Polyethylene (UHMWPE) and Ti-6Al-4V were used for the glenoid fossa part and condylar part of the TMJ Implant, respectively. In the second material model, Co-Cr alloy was used for the condylar part and UHMWPE for the glenoid fossa part of the TMJ Implant. For fasteners, it was decided to use Ti-Alloy screws as the perfect combination of material for the design of the patient-specific TMJ implant [7]. Owing to their excellent qualities, titanium and its alloys are regarded as the most ideal options for prostheses design [8]. Ti alloys with porosity are quickly becoming the preferred material for bio-implant uses [8–11].

The main aim of this article is to present a finite element study on patient-specific TMJ implants under external loading conditions and compare it with the original jawbone

loading condition and work on estimating von Mises stresses and deformations in the original jaw or repaired condylar head (with intact mandible), and in patient-specific TMJ implant using a finite element method in ANSYS software, and compare the results. Additionally, carrying out of finite element study for analysing the biomechanical behaviour of the patient-specific temporomandibular joint implant and to decide the specification that would lead to a more advantageous and safer implantation option during the operation.

## 2. Methodology

In this research work, a three-dimensional model of the mandible jaw was generated from DICOM files obtained with the courtesy of Oral and Maxillofacial surgeon Dr. Rajesh Dhirawani, Jabalpur Research Centre. That patient had an injury on his right condylar head during cricket practice. To perform this study, a CT scan was performed on the injured patient's skull. CT scan machine gives DICOM (Digital Imaging and Communications in Medicine) file output that is converted into Nrrd (Nearly raw raster data) file. Nrrd file extension plays an important role in processing images from CT scans. Afterward, Nrrd files were converted into STL file format using Democratiz3D. STL files format contains all data in 3D vector form where 3D geometry was formed by joining small triangles, which have coordinated location from origin. STL file acts as a bridge between scanned file and an editable CAD file, and the whole design was carried out with the reference of STL file. After obtaining STL files of the skull-mandibular region, extra bones and parts of the skull were removed from geometry as we only needed jaw, condylar part, and fossa part of skull bone. After removing extra geometry, geometry refinement was performed to obtain the better finish of geometry in the Meshmixer tool since the obtained 3D model is in a rough condition that cannot be used for further analysis; thereafter STL files were imported in SOLIDWORKS 2018, which is a mechanical CAD tool. In SOLIDWORKS, we repaired the injured right condylar head by mirroring the left condylar as shown in Figure 1a,b and then the geometry file was saved as an IGS file. Again, the STL file was imported to SOLIDWORKS to design a patient-specific TMJ Implant. References of condylar surface and fossa were taken to design an implant for an exact fit to the patient. CAD file of the custom implant was saved to IGES file format. Linear static analysis was performed on both designs, one is naturally repaired condylar and TMJ and the other is patient-specific TMJ implant.

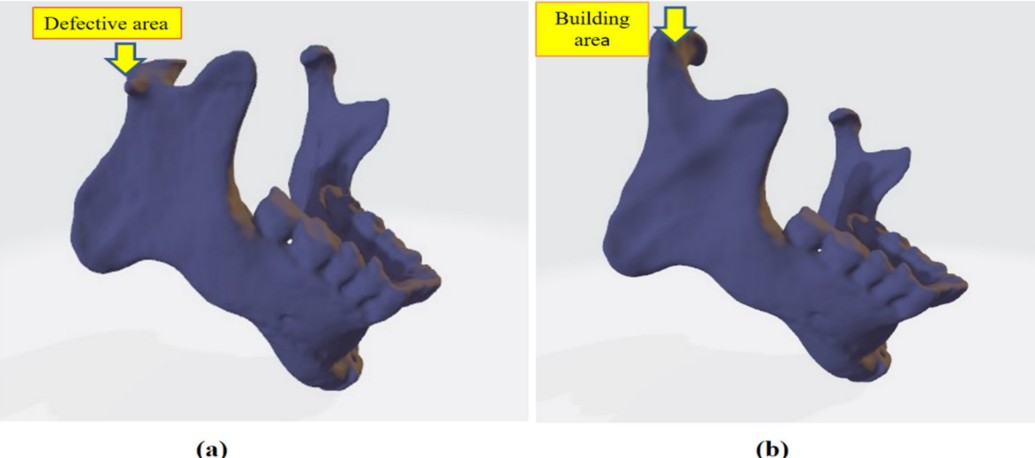

(a)          (b)

**Figure 1.** (**a**) The mandible jaw with damaged condyle, and (**b**) Later on rebuild model.

Afterward, for the generation of the mandible jaw of the patient, the file in IGES format was imported into the ANSYS workbench. Main steps concerning the design and numerical simulation of patient-specific TMJ Implant are represented in Figure 2.

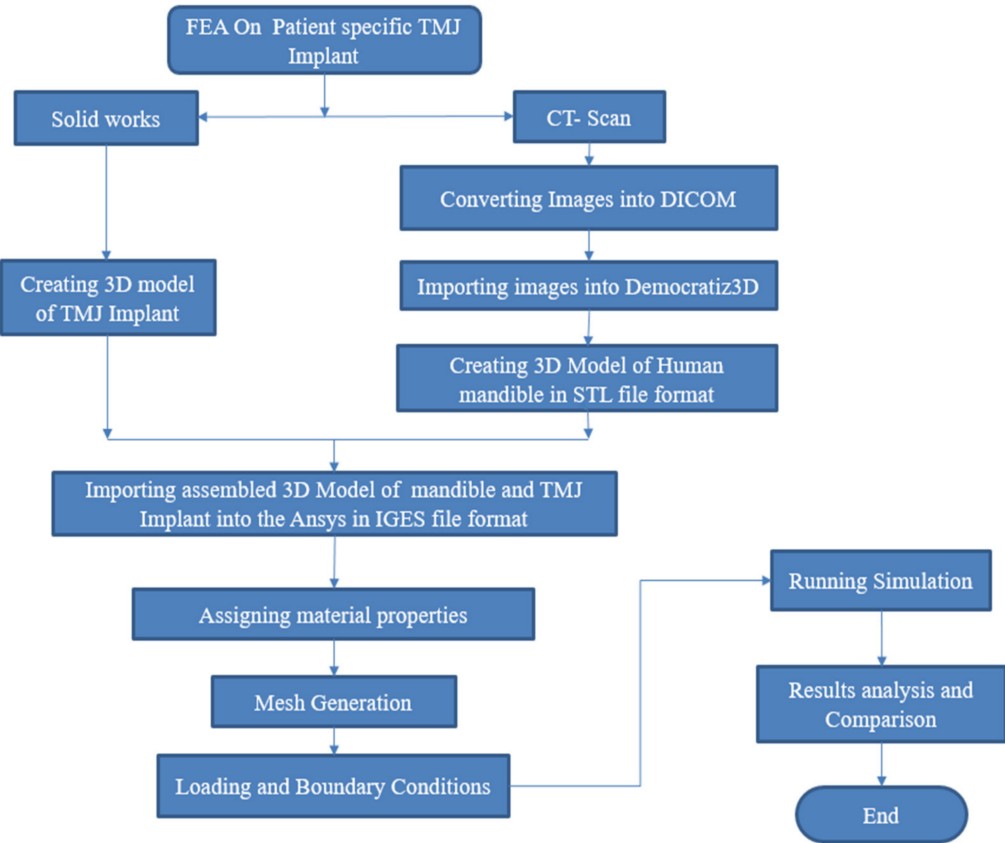

**Figure 2.** Flow chart of finite element analysis of patient-specific TMJ prosthesis.

*2.1. Materials*

Bone properties were given to the bone parts of the jawbone, glenoid fossa and articular eminence, and these all have been assumed to be homogeneous, isotropic and linear elastic [12–15]. Despite the fact that bone is an anisotropic material, earlier research by Daegling DJ et al. [16] and Ichim et al. [15,17] have shown that isotropic models of the mandible may detect considerable strain changes when simulating functional loads. Hence, in the current study the material was considered isotropic rather than orthotropic in nature. All anatomical components of the bone i.e., cortical, cancellous and teeth have been assigned the properties of cortical bone [12]. In the current study with the primary purpose to investigate the biomechanical behaviour of the condylar area where the implant was implanted, finite element analysis was carried out on the patient-specific TMJ implant which is normally made of the same material. Hence, the current study does not differentiate between the cortical and trabecular bones. In similar studies undertaken by Gregolin et al. [12], the same material properties were considered for the trabecular and cortical bone. Ichim et al. [13] and Liu et al. [14] demonstrated that the thickness of the cortical and trabecular bone has no significant influence on the distribution of stresses through the prosthetic components and the supporting bone. In the simulation study, for the human bone, Young's modulus of elasticity and Poisson's ratio 14.7 GPa and 0.3 have been used, respectively [6]. Co-Cr and Ti alloy (Ti-6Al-4V) material were used for combined condylar and ramus part of TMJ implant for separate analysis. UHMWPE material was assigned for the upper part of the implant (glenoid fossa and articular eminence) and titanium alloy was used for all screws in the TMJ Implant. All material properties used in FEA are shown in Table 1.

**Table 1.** Materials properties used in finite element analysis [6].

| Material | Young's Modulus MPa | Poisson's Ratio |
|---|---|---|
| UHMWPE | 500 | 0.29 |
| Ti-6Al-4V | $1.1 \times 10^5$ | 0.3 |
| Co-Cr alloy | $2.2 \times 10^5$ | 0.3 |
| Human Bone | $14.7 \times 10^3$ | 0.28 |

*2.2. Finite Element Analysis*

The customized TMJ Implant and intact natural mandible were both studied and analysed in ANSYS workbench R15 to study and analyse the biomechanical behaviour of the patient-specific TMJ prostheses implant for static loading scenario. Patient-specific total TMJ implant was designed using SOLIDWORKS 2018 designing software. The geometry of the model with dimensions are shown in Figure 3.

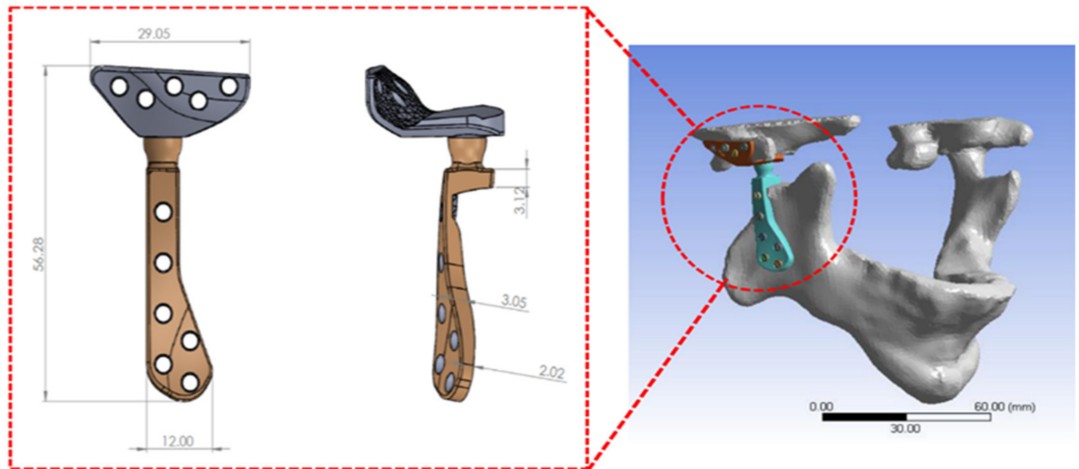

**Figure 3.** Solid works design of patient-specific TMJ implant with schematic of model in actual situation.

The designed solid model is then converted into a mesh model to perform static simulation. ANSYS (Workbench R15) Software was utilized to mesh the assembly using the 4-noded tetrahedral linear element with six degrees of freedom. Bonded contacts have been used for the interfaces between TMJ Implant and jaw-mandible. The numbers of nodes and elements of the model were 153,610 and 87,932 for the combined mandible and natural TMJ in the mesh model while corresponding numbers of nodes and elements for designed patient-specific implant were 166,512 and 94,653, respectively. The convergence of the solutions for both cases was obtained at the element size of 0.5 mm. The masticatory muscles have a remarkable impact on the temporomandibular joint assembly. These muscles are mainly involved in the transferring of chewing forces to the TMJ. In this study, we have considered the three muscles that are involved in mouth closing, because these muscles are always involved in the biting and chewing process which in turn exerts pressure on the TMJ complex. All loading conditions were given as per standard muscle loading magnitude and direction by temporal, pterygoid and masseter muscle. The masticatory muscles load of 1 kN was transferred and distributed uniformly to the mandible into the corresponding enclosures of the masseter, medial pterygoid and temporal jaw muscles. In this simulation study, the intense situation of chewing was used in order to authenticate the reaction of the mandible jaw to the severe situations of implementation of chewing forces [18]. Finally, the simulation was conducted on the model by keeping boundary conditions from the literature as shown below in Table 2 [19,20].

**Table 2.** Magnitude and direction of the muscle forces in the present work [21].

| Muscles | Magnitude (N) | X[p]-Axis (N) | Y[q]-Axis (N) | Z[r]-Axis (N) |
|---|---|---|---|---|
| Masseter right | 250 | 104.7 | −51.7 | 221.2 |
| Masseter left | 250 | −104.7 | −51.7 | 221.2 |
| Temporal right | 100 | −50 | 22.1 | 83.7 |
| Temporal left | 100 | 50 | 22.1 | 83.7 |
| Medial pterygoid R | 150 | 55.8 | −72.9 | 118.6 |
| Medial pterygoid L | 150 | −55.8 | −72.9 | 118.6 |

[p] The X axis is directed perpendicular to the sagittal plane. [q] The Y axis is directed perpendicular to the coronal plane. [r] The Z axis is directed perpendicular to the axial plane.

The masseter muscles, medial pterygoid muscles and the temporalis muscles were subjected to a load of 500 N, 300 N and 200 N, respectively, which in turn represents the muscle reaction force during the mastication. Table 2 displays the directions of muscle forces in the three-dimensional axis of the coordinate system. The upper portion of the articular fossa was approximated as rigidly fixed. One more approximation of high stiffness springs placed symmetrically in INCISOR teeth [22] was taken in the present study with all boundary conditions and constraints as depicted in Figure 4.

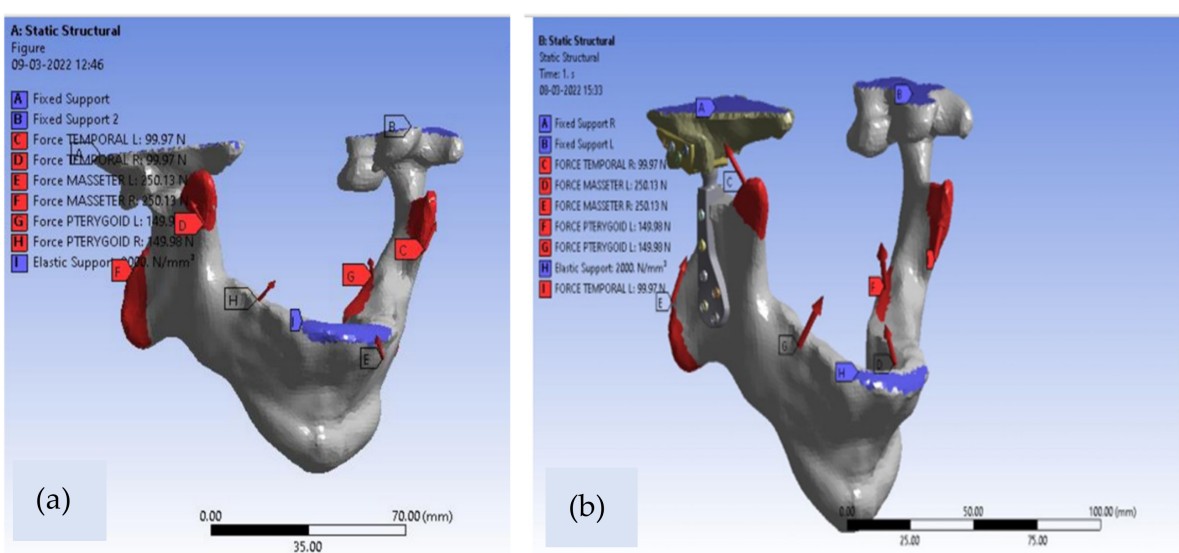

**Figure 4.** Schematic representation of model constraints (boundary conditions) and load application for FEA in (**a**) intact mandible and (**b**) customized TMJ Implant.

## 3. Results and Discussions

Von-Mises stresses and deformations contours for the intact mandible are depicted in Figure 5a,b. For the intact natural mandible, the occurred value of maximum von Mises stress is 14.55 MPa, which can be noticed in the inferior part of the collar of the condyle as clearly shown below in Figure 5a,b. The observed deformation near the condylar head is 0.45 mm.

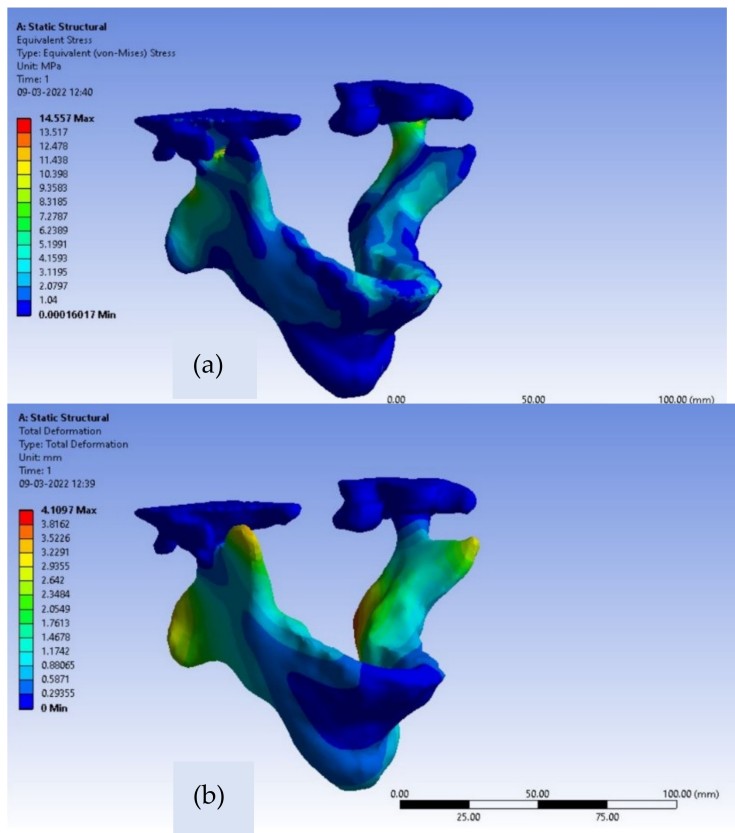

**Figure 5.** For intact mandible (**a**) The von-Mises stresses contours (**b**) deformations contours.

Moreover, the value of maximum stress, found at the inferior part of the collar of the condyle, is far behind the ultimate strength of the adult bone in the range of 130–190 MPa [20]. For customized total TMJ prostheses, the von-Mises stress contours for Ti alloy and Co-Cr alloy are shown in Figures 6 and 7, respectively.

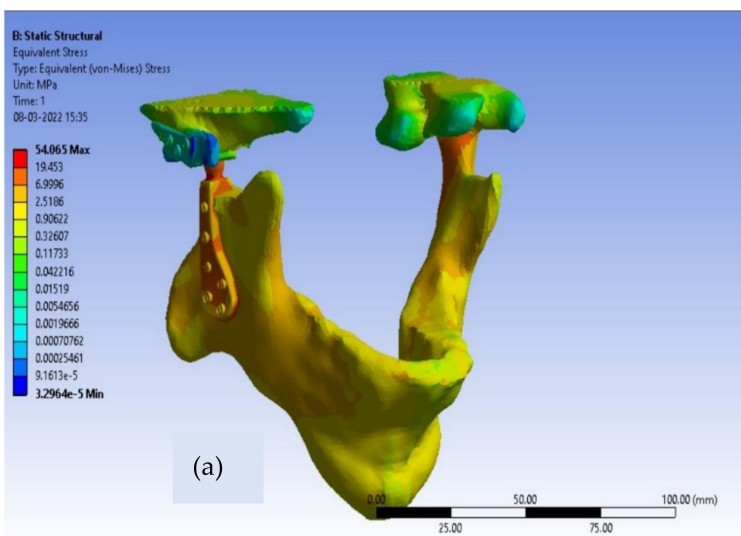

**Figure 6.** *Cont.*

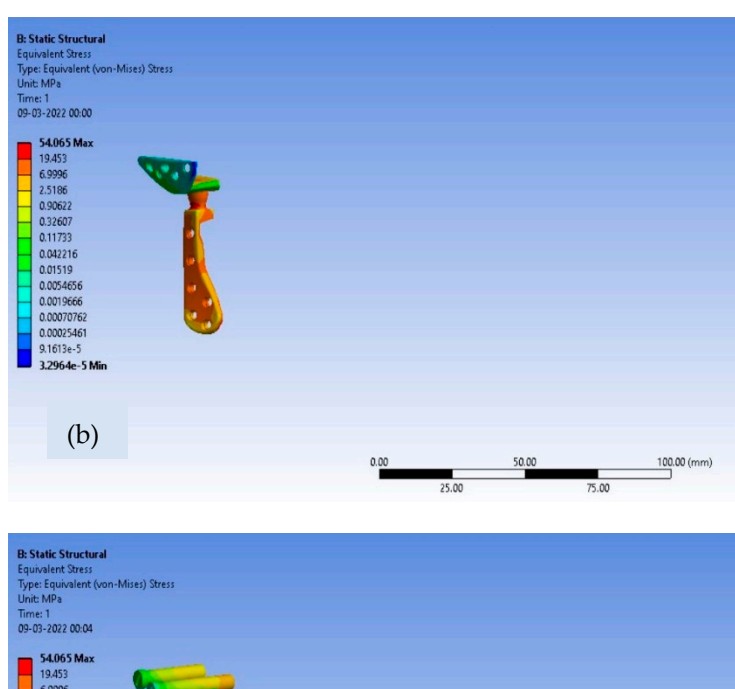

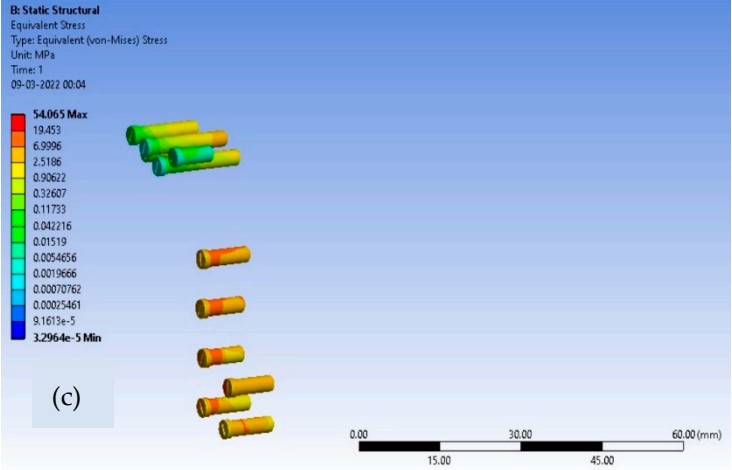

**Figure 6.** The von-Mises stress contours for (**a**) Total TMJ prostheses with mandible (**b**) Condylar and fossa parts without screws (**c**) fossa and condylar screws in case of Ti-alloy model of customized TMJ prostheses.

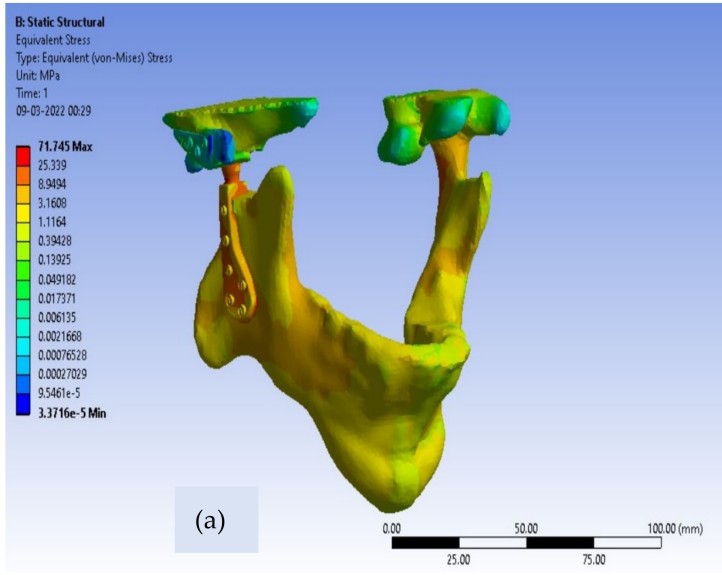

**Figure 7.** *Cont*.

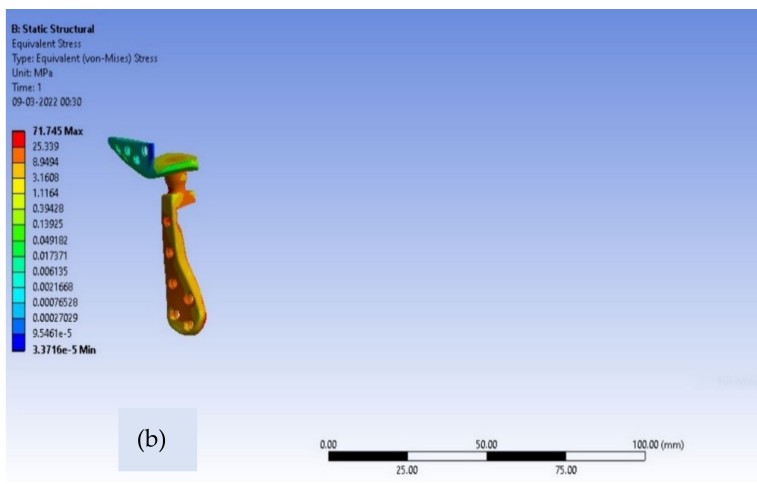

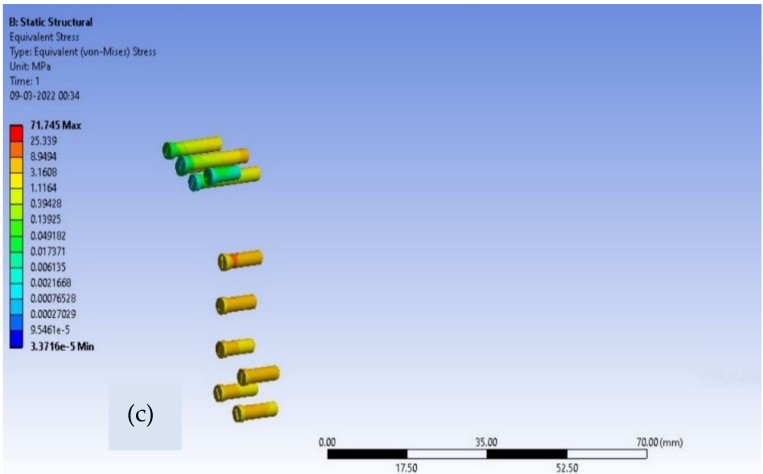

**Figure 7.** The von-Mises stress contours for (**a**) Total TMJ prostheses with mandible (**b**) Condylar and fossa parts without screws (**c**) fossa and condylar screws in case of Co-Cr alloy model of customized TMJ prostheses.

The greatest observed value of von-Mises stress near the neck of the condylar head in Ti alloy implant material is 32 MPa, as shown in Table 3, which is safe enough because the yield strength of Ti-6Al-4V is 850 MPa. The maximum produced von-Mises stresses around the condylar area for the Co-Cr alloy material model of Customized TMJ prostheses are 45 MPa, which is likewise substantially lower than the Co-Cr alloy yield strength (448 MPa). Therefore, we can conclude that both of the material models are in the safer zone as per the design point of view of patient-specific TMJ Implant. Von-Mises stresses for the condylar component were observed to be lower in Ti alloy-based customized TMJ implants than in Co-Cr alloy-based customized TMJ implants [23].

**Table 3.** Maximum von-Mises stresses values in MPa for two different materials model in customized TMJ prostheses.

| Material Model | Condyle | Condylar Screws | Fossa | Fossa Screws |
|:---:|:---:|:---:|:---:|:---:|
| Ti alloy | 32 | 30.48 | 7 | 18.28 |
| Co-Cr alloy | 45 | 31.62 | 8 | 22.3 |

For the fossa components corresponding to both material models, the differences in values of Von Mises stresses are very small. Further, for patient-specific TMJ prostheses, the deformation contours for Ti alloy and Co-Cr alloy are shown in Figures 8 and 9, respectively.

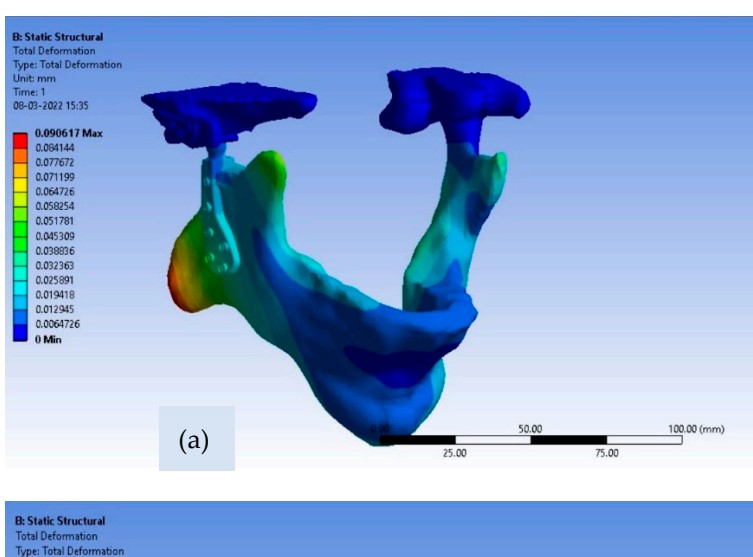

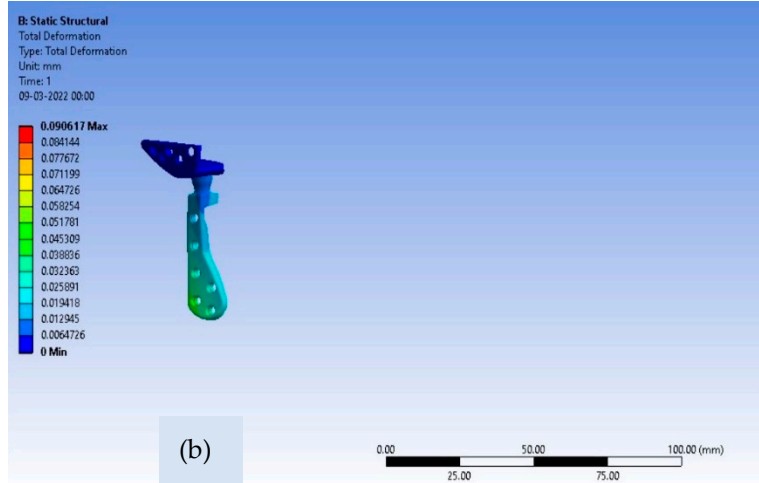

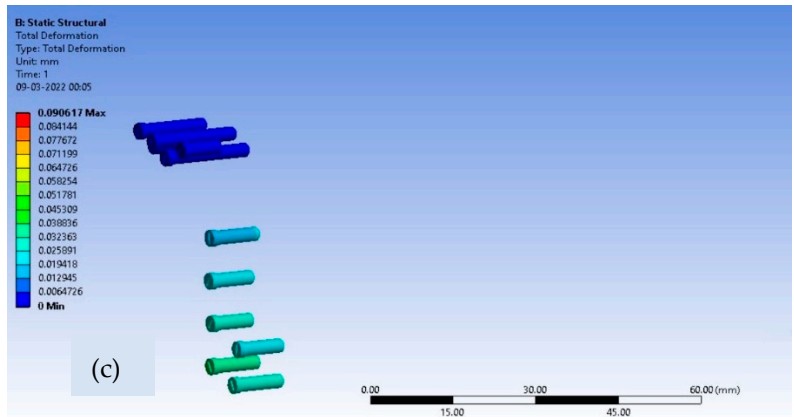

**Figure 8.** Deformation contours for (**a**) Total TMJ prostheses with mandible (**b**) Condylar and fossa parts without screws (**c**) fossa and condylar screws in case of Ti alloy model of customized TMJ prostheses.

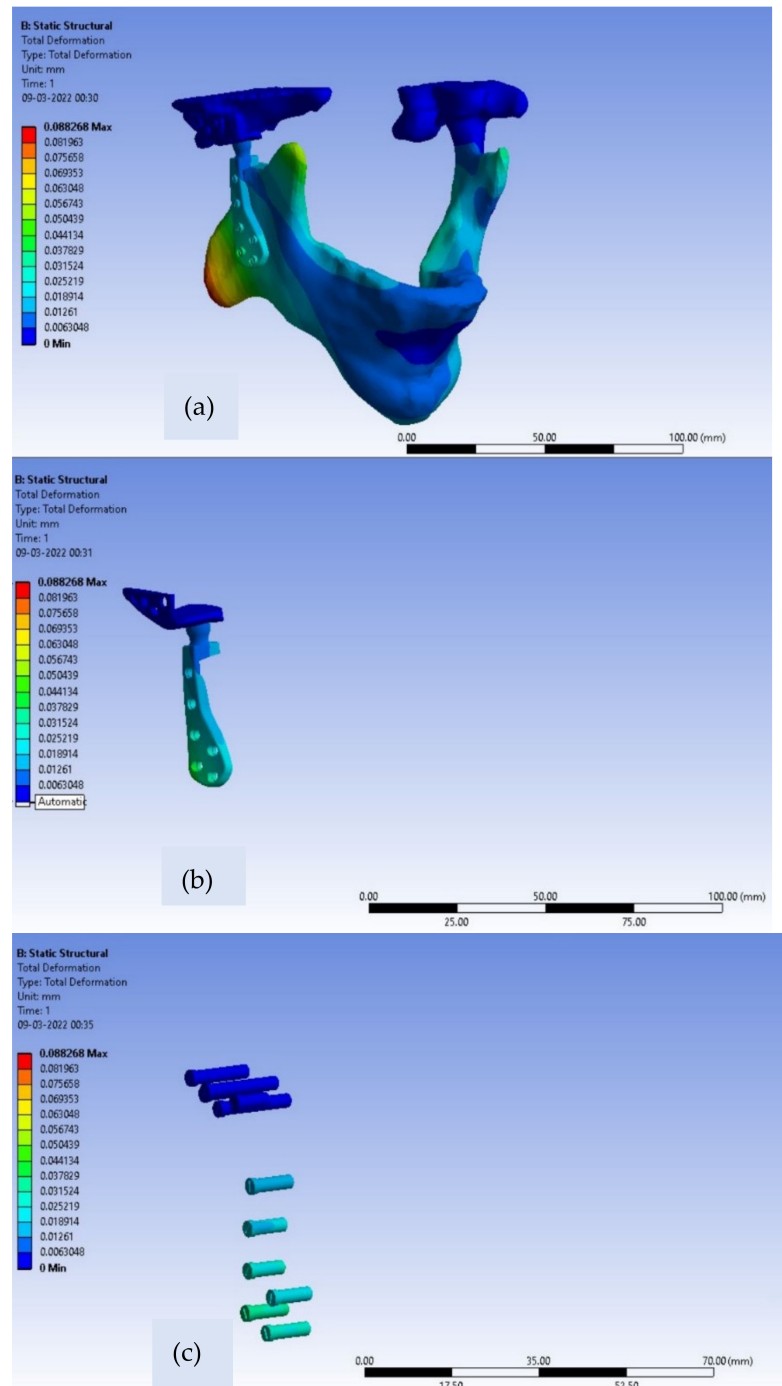

**Figure 9.** Deformation contours for (**a**) Total TMJ prostheses with mandible (**b**) Condylar and fossa parts without screws (**c**) fossa and condylar screws in case of Co-Cr alloy model of customized TMJ prostheses.

The maximum value of deformation for the Ti alloy model and Co-Cr model of customized TMJ prostheses are 0.022 mm and 0.0113 mm, respectively, shown below in Table 4.

**Table 4.** Maximum deformation values in mm for two different materials model in customized TMJ prostheses.

| Material Model | Condyle | Condylar Screws | Fossa | Fossa Screws |
|---|---|---|---|---|
| Model 1 [a] | 0.022 | 0.015 | 0.005 | 0.005 |
| Model 2 [b] | 0.0113 | 0.017 | 0.006 | 0.005 |

[a] Model 1 refers to Ti alloy; [b] Model 2 refers to Co-Cr alloy.

As compared to the intact mandible model, both material models of customized implant exhibit higher induced stresses and lower deformation values. The reason for this is that the Young's modulus of the jawbone is lower as compared with Ti alloy and Co-Cr alloy, therefore a flexible bone changes its shape considerably more than the other two material models when subjected to loading. Moreover, the less stiff the material, the less the resistance to the external load will be, hence the induced stress in the case of intact mandible model is lower than other two material models of customized TMJ implant. The TMJ implant must be adequately stiffer as mechanical failures can fracture the implant. The most critical safety requirements for bio-implants are mechanical strength and toughness. The TMJ Implant must be sturdy and durable enough to survive the physiological pressures placed on them, and they must be anticipated to last a long time or until they fail or require more than one surgical operation. Any prostheses or bio-implants must have a strength greater than the bone and an elastic modulus that is comparable to that of human bone to avoid stress shielding effects which in turn results in bone resorption [8]. Therefore, the outcome of our study is that our patient-specific TMJ prostheses are on the safer side from a designer's point of view. Similarly, since Ti alloy is less stiff than Co-Cr alloy; hence the maximum deformation of Ti alloy material model is greater and the maximum von-Mises stress is lesser than Co-Cr material model of customized TMJ prostheses. This observation also makes Ti-alloy (Ti-6Al-4V) a material that could be a better choice for the condylar and ramus part of the patient-specific implant from a design perspective. In this study of finite element analyses of patient-specific TMJ Implant, the highest stresses were always found nearer the collar of condyle.

The use of metal alloys with a lowered Young's modulus as a TMJ Implant holds a bright future as far as the stress shielding and weight reduction of implant are concerned. Due to the efficient stress transfer between the bone and the implant, low modulus alloys are very useful in minimizing bone resorption and promoting bone remodelling. To reduce the stress shielding effect, biomedical implants should have a modulus that matches that of human bone, but at the same time the implant should exhibit good fracture toughness or fatigue strength as well ensure a good lifespan of the implant with excellent biocompatibility.

For future work, dynamic analysis of patient-specific TMJ implants can be performed with the inclusion of a TMJ disk. Moreover, we can compare customized TMJ implants with stock TMJ prostheses.

## 4. Conclusions

Stress shielding and toxic effects represent some limitations for the usage of Co-Cr as a TMJ Implant. Despite the lesser fatigue strength compared to Co-Cr alloys, Ti alloys are popularly used for TMJ implants due to their extraordinary bio-compatibility. The modulus of elasticity of Ti alloys are much lower than the Co-Cr and stainless steel alloys, this renders Ti alloys more preferable over the Co-Cr and stainless steel alloys as far as stress shielding is concerned. But as compared to human bone, Ti alloys are still much stiffer which cannot completely eliminate stress shielding effects for long term use.

The β-titanium alloys with porosity contains huge potential for the development of temporomandibular joint (TMJ) prostheses. The scientific and research community are working in the direction to develop such materials for bio-medical implants.

The FEA of this study suggests that the patient-specific TMJ implant reveals lesser maximum deformation values near the neck of the condylar head as compared to the intact mandible. This implant design exhibits a better option of customized TMJ implant for a person suffering critically from TMD with an enhanced design. Further, in vivo and in vitro testing and verification are inevitably required to ensure viability and efficacy before putting customized TMJ prostheses into clinical use. This study can provide aid to any biomedical engineers and scientists to develop a patient-specific temporo-mandibular implant. Moreover, every human being bears a peculiar shape and geometry of their own mandible and jaw, which is why the FEA results of the present study exhibit some differences as compared to other concerned research works. This study provides useful information concerning the stress distribution and deformations for development of customized temporomandibular joint implants using finite element simulations with the combinations of different material models.

**Author Contributions:** A.T.: Conceptualization, Methodology, Software, Writing Original Draft, Visualization, Investigation, and Original draft preparation. V.K.G.: Conceptualization, Methodology, Supervision, Software, Validation, Visualization, and Validation of research methods. R.K.H.: software, formal analysis; writing—review and editing. I.A.P.: supervision; funding acquisition. All authors have read and agreed to the published version of the manuscript.

**Funding:** This research was funded by the grant from the Ministry of Science and Higher Education of Russia supported by Southern Federal University, grant No. VnGr-07/2020-04-IM.

**Institutional Review Board Statement:** Not applicable.

**Informed Consent Statement:** Not applicable.

**Acknowledgments:** This research was funded by the grant from the Ministry of Science and Higher Education of Russia supported by Southern Federal University, grant No. VnGr-07/2020-04-IM. With the experience and suggestions/feedbacks of Rajesh Dhirawani Maxillofacial Surgeon Jabalpur Research Centre, M.P, India. The author is grateful for the support from the Ministry of Human Resource Development, Government of India for the Institute Teaching Assistantship to Anubhav Tiwari for conducting this research work as a part of his Ph.D.

**Conflicts of Interest:** The authors declare no conflict of interest.

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
