# Peer review of "Biomechanical Analysis of Patient-Specific Temporomandibular Joint Implant and Comparison with Natural Intact Jaw Bone Using Finite Element Method"

_applsci, doi:10.3390/app12063003_

Round 1

Reviewer 1 Report

This paper conducted a finite element analysis comparison between TMJ implant and natural intact TMJ. The result is of interest for researchers in this field of study. However, bone model is considered obsolete. Additional study may improve the quality of the results. My comments are listed below.

  1. Jaw bone consist of cortical and trabecular bones which have different material properties. However, this study did not consider this. At least two material properties of bone should be used for reasonable analysis result.
  2. Figure 3: Please use muscles’ name shown in Table 2, instead of “force”.
  3. How did you construct the coordinate system? More information regarding the coordinate system is required.
  4. Figure 4 to 8: Please unify the scale of the contour fringe for each parameter, so it will be easier to compare.
  5. Table 4: What are Model 1 and Model 2? There was no explanation about these.
  6. Figures are unsharp, especially fonts and numbers inside the figures. Sharp and clear figures are recommended.

Author Response

'' Please see the attachment''

Reviewer 2 Report

This research paper addresses a current issue in maxillofacial surgery. Total alloplastic TMJ replacements are essential in the treatment of extreme pathologies of this joint. 
Kindly justify the use of isotropic rather than orthotropic modelling and why the authors believe that the TMJ implant must be stiffer than the prototype. And why it has to be many times stiffer. What do the authors think about metal alloys with lowered Young's modulus for these reconstructions which do not carry the weight of the whole body?
Please consider expanding the literature with works
Li Y, Yang C, Zhao H, Qu S, Li X, Li Y. New Developments of Ti-Based Alloys for Biomedical Applications. Materials (Basel). 2014 Mar 4;7(3):1709-1800. doi: 10.3390/ma7031709. PMID: 28788539; PMCID: PMC5453259.
Zhang E, Zhao X, Hu J, Wang R, Fu S, Qin G. Antibacterial metals and alloys for potential biomedical implants. Bioact Mater. 2021 Feb 8;6(8):2569-2612. doi: 10.1016/j.bioactmat.2021.01.030. PMID: 33615045; PMCID: PMC7876544.
Kim KM, Kim HY, Miyazaki S. Effect of Zr Content on Phase Stability, Deformation Behavior, and Young's Modulus in Ti-Nb-Zr Alloys. Materials (Basel). 2020 Jan 19;13(2):476. doi: 10.3390/ma13020476. PMID: 31963854; PMCID: PMC7014103.
Kozakiewicz M, Wach T, Szymor P, Zieliński R. Two different techniques of manufacturing TMJ replacements - A technical report. J Craniomaxillofac Surg. 2017 Sep;45(9):1432-1437. doi: 10.1016/j.jcms.2017.06.003. Epub 2017 Jun 12. PMID: 28687468.

Author Response

''Please see the attachment''

Round 2

Reviewer 1 Report

Thank you for your response.